# Effects of Both Fiber Post/Core Resin Construction System and Root Canal Sealer on the Material Interface in Deep Areas of Root Canal

**DOI:** 10.3390/ma14040982

**Published:** 2021-02-19

**Authors:** Hiroki Miura, Shinji Yoshii, Masataka Fujimoto, Ayako Washio, Takahiko Morotomi, Hiroshi Ikeda, Chiaki Kitamura

**Affiliations:** 1Division of Endodontics and Restorative Dentistry, Department of Oral Functions, Kyushu Dental University, 2-6-1 Manazuru, Kokurakita-ku, Kitakyushu, Fukuoka 803-8580, Japan; r17miura@fa.kyu-dent.ac.jp (H.M.); r08yoshii@fa.kyu-dent.ac.jp (S.Y.); r12fujimoto@fa.kyu-dent.ac.jp (M.F.); r05washio@fa.kyu-dent.ac.jp (A.W.); r13morotomi@fa.kyu-dent.ac.jp (T.M.); 2Division of Biomaterials, Department of Oral Functions, Kyushu Dental University, 2-6-1 Manazuru, Kokurakita-ku, Kitakyushu, Fukuoka 803-8580, Japan; r16ikeda@fa.kyu-dent.ac.jp

**Keywords:** fiber post core construction system, composite resin, root canal sealer

## Abstract

This study aimed to examine the resin polymerization of a fiber post/core resin construction system and the interface between resin and root canal sealers, which are important for root canal sealing. We used the i-TFC Luminus fiber post and i-TFC Luminus LC flow (i-TFC-L), the GC fiber post and Unifil Core EM (GCF), and the FiberKor post and Build-It FR (FKP) as core construction systems, and Nishika Canal Sealer BG (CS-BG), Metaseal Soft (META), and Nishika Canal Sealer EN (CS-EN) as sealers. The light transmission of fiber posts (n = 5), the polymerization of core resin (n = 5), and the adhesion between the sealer and core resin (n = 10) were evaluated. The i-TFC Luminus fiber post light transmission was significantly higher than that of other posts. Without shielding, i-TFC-L showed a significantly greater amount of polymerized resin than the other systems. With shielding, although i-TFC-L showed a significantly greater amount of polymerized resin immediately after light irradiation, polymerized resin was significantly greater in GCF and FKP after 10 min. All systems adhered to CS-BG and META but not to CS-EN. These results indicate that resin polymerization in the cavity differs among fiber post/core resin construction systems and that the adhesion of the resin and sealer depends on the property of the sealer.

## 1. Introduction

The sealing of endodontically treated teeth is affected by the material properties of the root canal filling and the fiber post/core resin construction system [1]. Endodontically treated teeth were previously commonly filled with gutta-percha and root canal sealer. Traditionally, eugenol-based sealers with antibacterial and sterilizing properties were used as major root canal sealers, but effective sealing ability in the root canal was not guaranteed because of its poor dentin adhesion. In the 2000s, a resin-based sealer containing 4-methacryloxyethyl trimellitic anhydride (4-META) was introduced for root canal wall adherence [2]. In recent years, bioceramics-based sealers with high biocompatibility that bond to the root canal wall by inducing hydroxyapatite formation have emerged [3,4,5]. In modern root canal fillings, resin- and bioceramics-based sealers play an important role in root canal sealing through adhesion/bond to the root canal wall [6,7,8,9].

Fiber posts, instead of metal posts, are now used in the restoration with composite resin because of their elastic moduli similar to dentin, resulting in a reduction in root fractures [10,11,12,13]. Polymerization of the composite resin in the deep areas of the root canal is important for successful core construction using fiber posts. However, fiber post structure was reported to affect the polymerization of composite resin in deep areas of the root canal [14]. The complete polymerization of composite resin for construction does not occur in the deep areas of the root canal [15,16,17,18], and the unpolymerized layer remains in the post cavity, even with the use of dual-cure-type composite resin for chemical polymerization [19,20]. Recently, a construction system combining a light-cure composite resin and fiber post with high translucency was developed and clinically applied to increase the resin polymerization rate in deep areas of the root canal [21,22,23,24,25]. Root canal filling materials and core construction materials have improved the sealing effectiveness in each development process. 

In clinical practice, the root canal sealer and core resin are used without considering the difference in the properties of each material. In addition, the adhesion of the fiber post/core resin construction system or root canal sealer to dentin has been reported extensively in previous research. However, there are few reports on the material interface between the composite resin used for construction and the sealer for the root canal filling, which is an important factor in achieving a root canal monoblock and complete root canal closure after core construction [26,27,28,29,30].

This study aimed to investigate the adhesion between the core resin of the construction system and the root canal sealer in the context of material property differences between the fiber post/core resin construction system and the root canal sealer. This study was based on two null hypotheses. The first null hypothesis was that the core resin of each core construction system polymerizes completely immediately after light irradiation. The second null hypothesis was that the core resin cannot adhere to each root canal sealer. We investigated the light transmittance of fiber posts, the extent of polymerization of construction composite resin, and the interface between the core resin and the root canal filling sealer in the deep areas of the root canal.

## 2. Materials and Methods

Three types of core construction systems (i-TFC Luminus fiber post and i-TFC Luminus Fiber LC Flow (i-TFC-L), Sun Medical Corp., Moriyama, Japan; GC fiber post N and GC Unifil Core EM (GCF), GC, Tokyo, Japan; and FiberKor post and Build-It FR (FKP), Pentron Corp., Wallingford, CT, USA); and three types of root canal sealers (Nishika Canal Sealer BG (CS-BG), Nippon Shika Yakuhin Co., Ltd., Shimonoseki, Japan; Metaseal Soft (META), Sun Medical Corp., Moriyama, Japan; and Nishika Canal Sealer EN (CS-EN), Nippon Shika Yakuhin Co., Ltd., Shimonoseki, Japan) were investigated (Table 1).

### 2.1. Fiber Post Transparency

Each fiber post was cut into 18 mm pieces and irradiated using a light irradiator (Radii Plus; SDI, Victoria, Australia) directly on the non-tapered side with the silicone shielding around the edge. The amount of light transmission (units: counts; specific wavelength: 459.5 nm) at the post apex through the fiber post (n = 5) was measured by a multi-channel spectroscope (FLAME-S-XR1-ES; OptoSirius, Tokyo, Japan) (Figure 1).

### 2.2. Determination of Degree of Polymerization

The polymerization of core resin in the root canal was determined by the weight change (%) before and after light irradiation. Regarding polymerization activation, the i-TFC-L core construction system involves photopolymerization, whereas the other two systems have a dual-cure mode.

A Teflon block with a semi-cylindrical cavity (diameter: 3 mm; depth: 15 mm) was fabricated as a root canal post cavity model (Figure 2a). After filling the cavity with core resin, a fiber post of the same manufacturer was inserted, and light was irradiated from the crown side. The light irradiation for 30 s was carried out with or without shielding around the fiber post at the entrance of the cavity. After 0, 5, and 10 min at 37 °C and 100% humidity, each specimen was isolated from the model then immersed into acetone to remove unpolymerized resin then washed and dried for 24 h. The weight of each specimen was measured using an electronic analytical balance (AUW220; Shimadzu, Kyoto, Japan) (Figure 2b).

Weight change was calculated according to the following Equation (1):Weight change(%) = (D − B)/(C − A − B) × 100(%),(1)
where A is the overall mold weight with or without the use of shielding, B is the fiber post weight, C is the total weight after light irradiation, and D is the specimen weight.

Polymerization depth was evaluated by measuring the length from the coronal end of the fiber post to the most apical point of the fiber post covered with hardened resin. A photograph and the measurement of the depth of polymerized resin are shown in Figure 3. Thirty samples were measured in these experiments (each group had 5 samples).

### 2.3. Interface of Core Resin and Root Canal Sealer

To analyze the interface of core resin and root canal sealer at the deep area of the root canal cavity, the shear bond test and microscopic analysis after the test were carried out. Figure 4 shows the schema of the experiment. The root canal sealer was hardened in the disk-shaped mold fabricated by acrylic resin (diameter: 10 mm; height: 2 mm). After one day, an artificial light blocking the root canal cavity model fabricated by the Teflon tube (inner diameter: 4 mm; height: 15 mm) was placed on the hardened root canal sealer and filled with core resin. After the injection of core resin into the tube, a silicone cover was used to block the light. A fiber post by the same manufacturer was inserted into the cavity and irradiated for 30 s and stored for 1 week. The shear bond strength was measured at a crosshead speed of 1.0 mm/min using a universal testing machine (AGS-H; Shimadzu, Kyoto, Japan). The prepared specimen was mounted along the horizontal axis adding shear strength along the vertical axis with 1.0 mm crosshead speed [31].

After the shear bond test, specimens were embedded with acrylic resin and cut vertically. The cut surface was polished to #8000 and observed under a scanning electron microscope (SEM) (JCM-7000, JEOL, Tokyo, Japan).

### 2.4. Statistical Analysis

The results of fiber post transparency, core resin weight change, and the shear bond test were analyzed using one-way analysis of variance (ANOVA) and Tukey’s test (*p* < 0.05). R software (Version R 3.5.0 GUI 1.70, 2016, The R Foundation, Vienna, Austria) was used to analyze.

## 3. Results

### 3.1. Light Transmittance of Fiber Posts

Table 2 shows each fiber post transparency. The i-TFC Luminus fiber showed significantly higher transparency values than the GC fiber post and FibreKor post. The FibreKor post had a significantly lower transparency value than the other posts.

### 3.2. Core Resin Weight Change

Table 3 shows the weight change in the core resin before and after light irradiation. When the upper part of the root canal was unshielded from light irradiation, the weight change for i-TFC-L (0 min: 97.5 ± 0.1%; 5 min: 96.5 ± 0.4%; 10 min: 96.6 ± 0.2%) was significantly higher than those of others. Regardless of the time after the light irradiation, the polymerization depth was 18.0 mm, which reached to the bottom of the cavity. GCF (0 min: 88.6 ± 2.0%; 5 min: 91.3 ± 2.4%) showed a significantly higher weight change than FKP (0 min: 44.7 ± 2.0%; 5 min: 90.1 ± 2.1%) at storage times of 0 and 5 min. At 0 min, the polymerization depth was 18.0 mm for GCF and 10.3 ± 0.4 mm for FKP. At 10 min, the weight change in FKP (96.6 ± 1.4%) was significantly higher than that in GCF (92.6 ± 2.3%), and there was no significant difference between FKP and i-TFC-L. After 5 min, the polymerization depth was 18.0 mm for both GCF and FKP.

When the upper part of the root canal was shielded from light, the weight change in i-TFC-L at 0 min was the highest (66.9 ± 4.7%), followed by GCF (41.5 ± 4.5%) and FKP (0.1 ± 0.1%). There was a significant difference among systems. The polymerization depth was 18.0, 11.3 ± 0.8, and 0 mm for i-TFC-L, GCF, and FKP, respectively. i-TFC-L showed the same weight change regardless of time after the irradiation (0 min: 66.9 ± 4.7%; 5 min: 72.7 ± 2.7%; 10 min: 72.2 ± 3.0%). At 10 min after the irradiation, GCF (82.9 ± 1.4%) and FKP (93.3 ± 1.4%) showed significantly higher values than i-TFC-L (72.2 ± 3.0%), and FKP showed more than 93%, regardless of shielding (96.6 ± 1.4% without shielding; 93.3 ± 1.4% with shielding).

At 5 and 10 min after the irradiation, the polymerization depth was 18.0 mm for all systems.

### 3.3. Interface of the Core Resin and Root Canal Sealer

Table 4 shows the results of the shear bond test. CS-BG and META adhered to all composite resins. The interface of core resin and CS-BG showed material fractures (n = 9) and interfacial fractures between the bonding material and CS-BG (n = 1). The interface of core resin and META showed material fractures occurring in all samples. No adhesion was observed in the interface of core resin and CS-EN.

Figure 5 shows representative results of the interface microstructure, showing adhesion of CS-BG and META to the core resin via a bonding layer.

## 4. Discussion

In a core construction system using a fiber post and core resin, the core resin contacts the root canal filling material in the deep area of the root canal. Recently, single-point root canal obturation using a resin- or bioceramics-based root canal sealer has received acceptance [6,7,8,9]. In this obturation, the main material in the root canal is root canal sealer, so the core resin mainly contacts to the root canal sealer. In the present study, the light transmittance of fiber posts, the polymerization of core resin, and the interface state between the core resin and root canal sealer were analyzed to clarify the effects of material properties of the fiber post/core resin construction system and the root canal sealer on the material interface in the deep areas of the root canal using three types of core construction system with different fiber post structures and composite resin polymerization modes and three types of root canal sealers with different compositions. A Teflon block with a semi-cylindrical cavity (diameter: 3 mm; depth: 15 mm) was fabricated and used as a root canal post cavity model in this study. Although various molds were used in the preliminary experiments, a Teflon block with a semi-cylindrical cavity was the easiest to separate from the hardened resin in the mold without breaking the sample.

First, the light transmittance of the fiber post and the core resin polymerization of each system were examined. The examination of light transmittance showed that the transparency value of i-TFC Luminus fiber was the highest and that of FiberKor post was the lowest. For the examination of core resin polymerization, light irradiation was carried out with or without the shielding around the fiber post at the entrance of the cavity to avoid direct irradiation of the resin. In this study, the degree of conversion (DC) in the core resin after the irradiation was not measured to analyze the amount of polymerized resin. It is known that measuring devices such as FTIR can evaluate DC, but it measures only the outermost surface, not the total amount. Therefore, hardened resin that remained after acetone immersion to remove uncured resin was considered as polymerized resin, and the weight change (%) before and after acetone immersion was used to estimate core resin polymerization. Polymerization of the light-cure-type composite resin of i-TFC-L was different in the presence or absence of shielding, whereas polymerization of both dual-cure-type composite resins increased, regardless of shielding, in a time-dependent manner.

Without shielding, core resin polymerization of i-TFC-L was greater than the other systems for all periods. The fiber post of i-TFC-L consists of optical fiber in the center and glass fiber that covers the optical fiber. In this fiber post, irradiated light is scattered laterally via the glass fiber. The highest core resin polymerization without shielding may be the result of photopolymerization of the light-cure-type composite resin, directly and indirectly via the light-transmitting fiber post.

With shielding, core resin polymerization of i-TFC-L was highest immediately after light irradiation, but after the irradiation the resin polymerization of i-TFC-L did not increase. We found that 10 min after the light irradiation, the polymerization of both GCF and FKP was higher than that of i-TFC-L. Differences in the progress of core resin polymerization may have resulted from material property differences of both fiber posts and core resin polymerization among the three systems. With shielding, the light scattered via the glass fiber of i-TFC Luminus fiber may have accelerated the photopolymerization of the light-cured composite resin only during irradiation, after which the polymerization stopped. On the other hand, the chemical polymerization of both GCF and FKP may have proceeded after the irradiation. Interestingly, GCF showed higher polymerization than FKP immediately after the irradiation; however, 10 min after the irradiation, the polymerization of FKP was greater than GCF. The light transmission of the FibreKor post was very low, and photopolymerization was not accelerated with shielding. Chemical polymerization of FKP core resin may proceed even after irradiation. The fiber post of GCF had light transmission, but it was less than half of that of i-TFC-L. According to previous studies, complete polymerization of the composite resin used for construction does not occur in the deep areas of the root canal. Further, these studies indicate that insufficient light irradiance for the polymerization of the dual-cure-type composite resin prevents sufficient curing reaction, reduces viscosity, and hinders radical transfer, ultimately preventing chemical polymerization and resulting in an unpolymerized layer [15,16,17,18,32]. The present results are consistent with those of previous reports and may indicate that the insufficient light passing through the GCF fiber post can interfere with the chemical polymerization of the core resin in the deep areas of the canal cavity model.

The results of polymerization depth showed that the extent of core resin polymerization finally reached the bottom of the post cavity in all fiber post/core resin construction systems, but the time taken for this was different between i-TFC-L and the other materials. There was polymerization of the i-TFC-L core resin to the bottom of the post cavity 0 min after light irradiation, whereas other materials needed 5 min or more, suggesting that sufficient storage time is necessary after light irradiation when fiber post core construction systems include dual-cure-type composite resin. Based on our results, the first null hypothesis that the core resin of each core construction system completely polymerizes immediately after light irradiation was rejected.

Next, the interface of core resin and root canal sealer was analyzed. In this study, three root canal sealers were used. CS-BG is bioceramics-based, META is resin-based, and CS-EN is eugenol-based. The shear bond test between the core resin and the root canal sealers showed values of 0.2–0.22 and 4–4.66 MPa for CS-BG and META, respectively. CS-EN did not adhere to any of the core resins. Regarding CS-EN, eugenol remained on the surface of the cured product, which inhibited the polymerization of the composite resin and prevented adherence [33]. Furthermore, SEM observation of the material interface for CS-BG and META revealed adhesion via a bonding layer with all composite resins. The calcium ions of CS-BG and the acidic monomer of the bonding agent may have bonded via a chemical reaction, and resins may have bonded to each other for META. Therefore, the second null hypothesis that the core resin cannot adhere to each root canal sealer was rejected. Recently, it was reported that the interface between composite resin and calcium silicate-based cements showed enough shear bond strength at several restoration timings, suggesting that calcium silicate-based cements may allow restorative procedures with both immediate and delayed timing [34]. The results of the present study using CS-BG are consistent with the result of immediate polymerization. We are now trying to clarify the shear bond strength of the interface between core resin and bioceramics-based sealers during several time frames, including delayed timing.

Overall, the present study indicates that the core resin polymerization of fiber post/resin core construction systems in the root canal cavity is affected by the light transmittance of the fiber post and the polymerization type of core resin, suggesting that it is necessary to consider the properties of each material when the fiber post core construction system is clinically used. Furthermore, regardless of resin polymerization type, the core resin of all fiber post core construction systems adhered to the bioceramics-based and resin-based canal sealers, but not the eugenol-based sealer, suggesting that the combination of root canal sealer and fiber post/core resin construction system is important to obtain the adhesion at the interface of materials and that the use of bioceramics-based or resin-based canal sealer may be essential for the establishment of root canal sealing. A limitation of this study is that the experiments were conducted in vitro and not on human teeth. Our future work will involve investigation of similar events in extracted human teeth and in vivo.

## 5. Conclusions

Within the limitations of this study, core resin polymerization in the root canal cavity differs among fiber post/core resin construction systems. In addition, adhesion of core resin and root canal sealer depends on the properties of the sealer. The use of bioceramics-based or resin-based root canal sealers that adhere to core resin is essential for root canal sealing after core construction.

## Figures and Tables

**Figure 1 materials-14-00982-f001:**
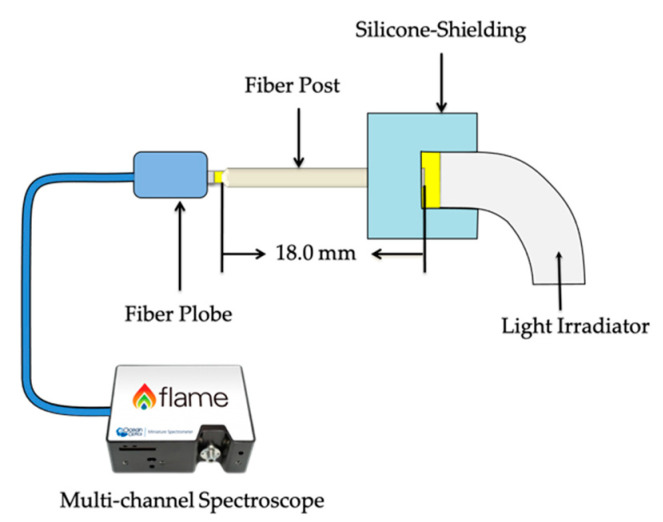
Schema of fiber post transmission test.

**Figure 2 materials-14-00982-f002:**
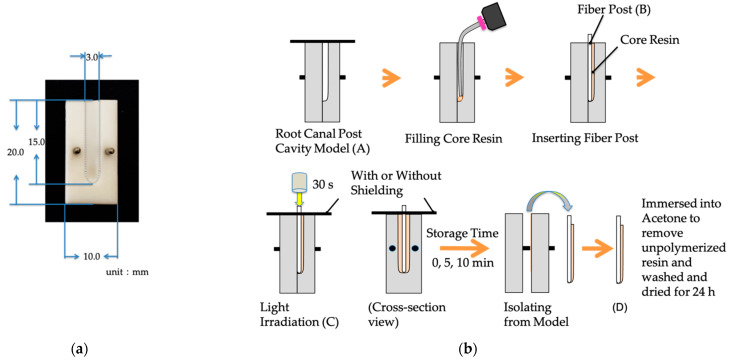
(**a**) The mold of the root canal cavity model. (**b**) Method of sample preparation.

**Figure 3 materials-14-00982-f003:**
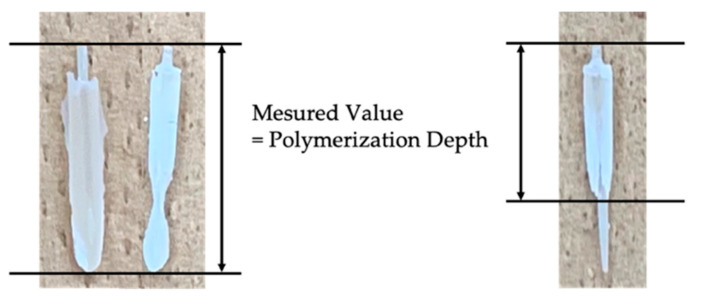
Polymerization depth.

**Figure 4 materials-14-00982-f004:**
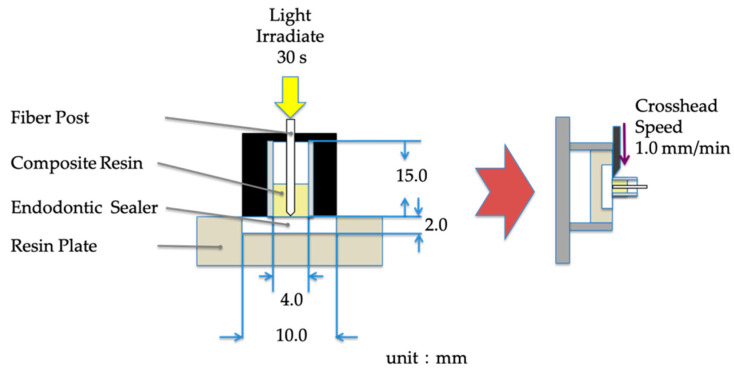
Schema of the experiment sample.

**Figure 5 materials-14-00982-f005:**
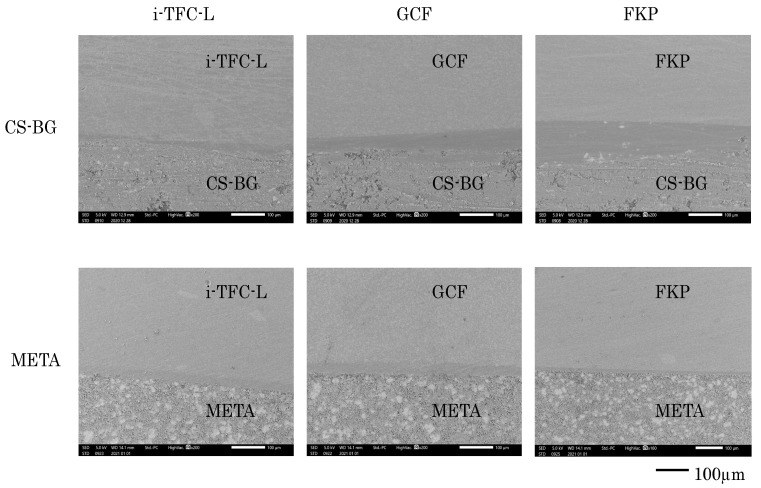
SEM micrographs of the surfaces of interface between core resin and root canal sealer (200×). Adhesive surface is observed in CS-BG group and META group regardless of the type of core resin.

**Table 1 materials-14-00982-t001:** Investigated materials. Bis-MPEPP:2,2-Bis[4-(methacryloxyethoxy)phenyl]propane, 4-META:4-methacryloxyethyl trimellitate anhydride, Bis-GMA: bisphenol A-glycidyl dimethacrylate, UDMA: urethane dimethacrylate, 4-MET: 4-methacryloxyethyl trimellitic acid, HDDMA: 1,6-hexanediol dimethacrylate, HEMA: hydroxyethyl methacrylate, MDP: 10-methacryloyloxydecyl dihydrogen phosphate.

	Manufacturer	Shape	Component	Lot	Code
i-TFC Luminus fiber	Sun medical Corp., Moriyama, Japan	φ 1.0 mm/tapered	Borosilicate glass, Barium oxide, Dimethacrylate and diacrylate copolymer	TW12	i-TFC-L
i-TFC Luminus Core LC flow			Bis-MPEPP, Barium Silica Glass, Aromatic diol methacrylate	RW4	
i-TFC Luminus bond			Bond: 4-META, Acetone, Water Catalyst: Aromatic amine, Aromatic sulfinate	RW13	
GC fiber post N	GC Corp., Tokyo, Japan	φ 1.0 mm/tapered	Slicate glass, Copolymer of methacryliate and Bis-GMA	2001291	GCF
GC Unifil core EM			Fluoro-aluminosilicate glass, UDMA	1809041	
Self-etching bond A&B			4-MET, methacrlate, ethanol, water	1809041	
FibreKor post	Pentron Corp., Wallingford, CT, USA	φ 1.0 mm/straight	Glass fiber, filler, Bis-GMA, HDDMA, UDMA	7537776	FKP
Build-It^TM^ FR			Bis-GMA, UDMA, HDDMA, barium borosilicate, Silica, Silane, Camphor quinone, Benzoyl peroxide	7558119	
E-Lize dentin bond II			HEMA, Bis-GMA, MDP, Silica, ethanol	190031	
Canal sealer BG	Nippon Shika Yakuhin Co., Ltd., Shimonoseki, Japan		Bioactive glass, Fatty acid, Bismuth subcarbonate, others	K36	CS-BG
Metaseal soft	Sun medical Corp., Moriyama, Japan		Powder:radiopaque filler, organic filler, hydrophilic chemical initiator Liquid:4-META, HEMA, di-methacrylates, water, photo-initiator	Powder:RM1 Liquid: SX1	META
Canal sealer E-N	Nippon Shika Yakuhin Co., Ltd., Shimonoseki, Japan		Eugenol, Rosin, Zinc oxide, Bismuth subcarbonate, others	K2F	CS-EN

**Table 2 materials-14-00982-t002:** Light transmittance test.

Fiber Post	i-TFC Luminus Fiber	GC Fiber Post	FibreKor Post
Counts	5195 ± 639 ^a^	2564 ± 667 ^b^	381 ± 11 ^c^

Different superscript letters indicate statistical differences in row (n = 5, *p* < 0.05).

**Table 3 materials-14-00982-t003:** Weight change.

**Without Shielding**			
**Storage Time**	**i-TFC-L**	**GCF**	**FKP**
0 min	97.5 ± 0.1 ^Aa^	88.6 ± 2.0 ^Ba^	44.7 ± 2.0 ^Ca^
5 min	96.5 ± 0.4 ^Aa^	91.3 ± 2.4 ^Bab^	90.1 ± 2.1 ^Cb^
10 min	96.6 ± 0.2 ^Aa^	92.6 ± 2.3 ^Bb^	96.6 ± 1.4 ^Ac^
**With Shielding**			
**Storage Time**	**i-TFC-L**	**GCF**	**FKP**
0 min	66.9 ± 4.7 ^Aa^	41.5 ± 4.5 ^Ba^	0.1 ± 0.1 ^Ca^
5 min	72.7 ± 2.7 ^Aa^	76.4 ± 4.1 ^Ab^	91.3 ± 2.0 ^Bb^
10 min	72.2 ± 3.0 ^Aa^	82.9 ± 1.4 ^Bc^	93.3 ± 1.4 ^Cb^

Same superscript capital letters indicate no significant differences (rows) for storage time. Same superscript lower case letters indicate no significant differences between each post materials (columns); (n = 5, *p* < 0.05).

**Table 4 materials-14-00982-t004:** The shear bond strength between core resin and root canal sealer.

Core Resin	Sealer	MPa (Ave)	SD
i-TFC-L	CS-BG	0.22 ^b^	0.05
	META	4.66 ^a^	1.11
	CS-EN	0	0
GCF	CS-BG	0.21 ^b^	0.04
	META	4 ^a^	0.99
	CS-EN	0	0
FKP	CS-BG	0.2 ^b^	0.06
	META	4.41 ^a^	1.06
	CS-EN	0	0

Different superscript letters indicate statistical differences in column (n = 10, *p* < 0.05).

## Data Availability

The date presented in this study are available on request from the corresponding author.

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
