# Peer review of "Effects of Both Fiber Post/Core Resin Construction System and Root Canal Sealer on the Material Interface in Deep Areas of Root Canal"

_materials, 2021, doi:10.3390/ma14040982_

Round 1

Reviewer 1 Report

The topic of the paper is quite interesting; materials and methods are well described, but I think there are some points that should be improved:

It is not clear to me how did you calculate the polymerization depth. How did you obtain "The length of polymerized resin"?

I suggest adding some schematic pictures to materials and methods, In order to easier the reading of the text, especially on page 2 line 70, and line 77.

The discussion session should be completely rewritten; currently, it is only a description of the results,  it is totally missing data interpretations and comparison with the previous literature.

How do you explain that with shielding, immediately after light irradiation,  i-TFC-L showed a significantly greater amount of polymerized but after 10 min, the polymerized resin was significantly greater in GCF and FKP than in i-TFC-L?

What are the strengths and limitations of this study?

Author Response

We provide a point-by-point response to the reviewer’s comments.

The modified and newly added text has been highlighted in yellow in the revised manuscript.

Reviewer 2 Report

Abstract

It must be reformulated, following the indications for the realization of the abstract, without the words: background, methods and results, and conclusion

Introduction

“In recent years, bioceramic-based sealers with high biocompatibility that bond to the root canal wall by inducing hydroxyapatite formation have emerged.” to support this sentence need an adequate reference, please read:  PMID: 31546696.

Please clarify the objective of this study and add the null hypothesis

Material and Methods:

  • Why was notISO  10993-12:2007 used which described sample preparation and reference materials.
  • Please specify the sample preparation; e.g. the light curing procedures (LCU, intensity, time, distance), covered with a foil or not; storage at which rel. humindity?
  • this reviewer still expresses concerns related to the statistical test method (parametric test for n=5). An independent and professional statistical evaluation is strongly recommended.
  • Number of samples. The authors should be clarifying this in each experiment.

Results

  • Lack of significance (p <???) in the explanations, in the results section. For example: “The Fi-125 berkor post had a significantly lower transparency value than the other posts”. Please add p-value

Discussion

  • The discussion was superficial and did not discuss the subject sufficiently. It could not adequately reveal the problem. What are the reasons for using theses tests, and what did they have advantages? These tests highlighted what. How can we interpret the results of these tests as a whole?
  • It was not explained how the this study will make a significant contribution to the study topic field.
  • Limitations?
  • Conclusions were not totally supported by the data showed.
  • Figure legends: Bad descriptions

Conclusions

  • Conclusions were not totally supported by the data showed.

References

  • Check reference’s format MDPI in the manuscript, and in the references. The article's titles of references have different formats are in different formats.

Author Response

(The authors gave the same response as above.)

Reviewer 3 Report

This research is under the scope of this journal; the topic is relevant for readers, and this research deals with potentially significant knowledge to the field.

However, there are some concerns in the about the present manuscript:

Abstract

  • It must be reformulated, following the indications for the realization of the abstract, without the words: background, methods and results and conclusion.
  • How many samples? Identified in the abstract.
  • In the results, is important to show more information, add some the p-values.

Introduction

  • Page 1 line 40, “high biocompatibility”, to support this sentence need an animal study reference, please read https://doi.org/10.3390/biomedicines9010024, also used this for the discussion.
  • What is the importance of this review for the clinical? You do not think this study are included to the others already done? Which results are comparable?
  • What this study has new?
  • What is the novelty of this paper? Please clarify in the appropriate section.
  • What was your hypothesis null hypothesis?

Materials and Methods

  • How was the sample calculated? Did authors perform a power analysis to evaluate if this sample size was appropriate?
  • When mentioning materials or devices: for some of them you don't mention the manufacturer at all, for some you mention only the manufacturer, for some the manufacturer and city, for some you mention the manufacturer and city/ country.

Results

  • Table 2 change “5,195 ± 639” to “5.195 ± 639” and “2,564 ± 667” to “2.564 ± 667”
  • Figure 2 need to improve size and quality resolution.

Discussion

  • Can did we restored the tooth same session them canal obturation? The authors need to discuss, Does delayed restoration improve bond strength to calcium silicate-based cement? Please read this article https://doi.org/10.1007/s00784-020-03640-7
  • Please, clarified What hat was the limitations of this study? And also, future perspectives.

References

  • were current and relevant and well inserted in the manuscript, just need a few more references, see the suggestion.

Author Response

(The authors gave the same response as above.)

Reviewer 4 Report

The Reference style is incorrect. Please read the author guidelines and proceed to the proper changes.you

Materials and Methods section

2.1. Fiber Post Transparency

Refer to the specific wavelength (nm) used for the current measurements.

2.2. Core Resin Weight Change

- Given that such a method aims to the gravimetric determination of the degree of double bond conversion of monomers during the polymerization process, the subtitle "Core Resin Weight Change" should be replaced by "Determination of degree of polymerization ".

-The experimental model described as "A Teflon block with a semi-cylindrical cavity (diameter: 3 mm; depth: 15 mm) was made as a root canal post cavity model" is associated with any standard method? How did authors decide to follow the above technique? Please explain your choice.

-The equation (1) should be rather written as follows:

-How many replicates per material were measured?

-"The length of polymerized resin along the fiber post vertical axis from the crown side edge was also measured as the polymerization depth." How did authors measure that distance? Please provide details.

2.3. Interface of Core Resin and Root Canal Sealer

- What was the material of the disk-shaped mold used for the root canal sealer preparation and of the artificial light blocking root canal cavity model? Did the authors refer to any standard method? Please provide details.

- How did authors mount the prepared specimens on the universal testing machine to measure the shear bond strength? Please provide details.

Thank you

Author Response

(The authors gave the same response as above.)

Round 2

Reviewer 1 Report

The authors met all my requirements of the first revision, but please pay attention to formatting the style of the references section.

Lines 199-221 should be further summarized, in no more than 3 lines

Author Response

We provide additional point-by-point response to the reviewer’s comments.

Reviewer 2 Report

Accept

Author Response

We appreciate your advice and your acceptance.

Reviewer 3 Report

This research is under the scope of this journal; the topic is interesting for readers and this research deals with potentially significant knowledge to the field and an open new way for future studies.

The authors improved the quality of the manuscript after the reviewer's indications.

Author Response

(The authors gave the same response as above.)

Reviewer 4 Report

Dear authors,

After the last consideration of both the revised form of the manuscript and your response letter to my suggestions, I would recommend the acceptance of your paper.

Good luck

Author Response

(The authors gave the same response as above.)
